# Determination of Plasmalogen Molecular Species in Hen Eggs

**DOI:** 10.3390/molecules29204795

**Published:** 2024-10-10

**Authors:** Taiki Miyazawa, Ohki Higuchi, Ryosuke Sogame, Teruo Miyazawa

**Affiliations:** 1New Industry Creation Hatchery Center (NICHe), Tohoku University, Sendai 980-8579, Miyagi, Japan; higuchi@hokkaido-bpi.co.jp (O.H.); sogame.ryosuke.p5@dc.tohoku.ac.jp (R.S.); 2Biodynamic Plant Institute Co., Ltd., Sapporo Techno Park, Sapporo 004-0015, Hokkaido, Japan

**Keywords:** Alzheimer’s disease, chicken egg, dementia, dry weight, docosahexaenoic acid, nutritional value, phosphatidylethanolamine, phospholipid, plasmalogen, tandem mass spectrometry

## Abstract

(1) Background: Plasmalogens are vinyl ether-type glycerophospholipids that are characteristically distributed in neural tissues and are significantly reduced in the brains of individuals with dementia compared to those in healthy subjects, suggesting a link between plasmalogen deficiency and cognitive decline. Hen eggs are expected to be a potential source of dietary plasmalogens, but the details remain unclear. (2) Methods: We evaluated the fresh weight, dry weight, total lipid, neutral lipids, glycolipids, and phospholipids in the egg yolk and egg white of hen egg. Then, the molecular species of plasmalogens were quantified using HPLC-ESI-MS/MS. (3) Results: In egg yolk, the total plasmalogen content was 1292.1 µg/100 g fresh weight and predominantly ethanolamine plasmalogens (PE-Pls), specifically 18:0/22:6-PE-Pls, which made up 75.6 wt% of the total plasmalogen. In egg white, the plasmalogen content was 31.4 µg/100 g fresh weight and predominantly PE-Pls, specifically 18:0/20:4-PE-Pls, which made up 49.6 wt% of the total plasmalogen. (4) Conclusions: Plasmalogens were found to be more enriched in egg yolk than in egg white. It was found that humans are likely to ingest almost 0.3 mg of total plasmalogens from one hen egg. These findings highlight the importance of plasmalogens in the daily diet, and it is recommended to explore the impact of long-term dietary plasmalogen intake to assess its effect on human health. This provides a viewpoint for the development of new food products.

## 1. Introduction

The global population is aging. According to a 2022 report by the World Health Organization (WHO), one in six people worldwide will be 60 years or older by 2030 [1]. In addition, the global population aged 60 and older is expected to double to 2.1 billion by 2050, and people 80 and older are expected to triple to 426 million [1]. This demographic shift is accompanied by a significant increase in the number of individuals with dementia, which is emerging as a major societal issue. According to a WHO report in 2021, the number of people with dementia, which stood at 55.2 million in 2017, is expected to rise to 78 million by 2030 and 139 million by 2050 [2]. In addition, a report released in 2022 by the Global Prevalence of Dementia (GBD) 2019 Dementia Forecasting Collaborators predicted that the number of worldwide dementia patients will triple between 2019 and 2050 [3]. These predictions indicate an urgent requirement for strategies to address the dementia care problem. One approach towards this need is to elucidate the impact of the daily intake of nutrients on cognitive function and use these findings to prevent or delay the progression of dementia [4].

Plasmalogens are a class of vinyl ether-type glycerophospholipids that are characteristically distributed in neural tissue and are significantly decreased in the brains and plasma of individuals with dementia compared to healthy subjects [5,6,7,8]. Among them, ethanolamine plasmalogens (PE-Pls) have been found as potential regulators of stress-induced neuronal cell death. In particular, docosahexaenoic acid (DHA)-containing PE-Pls (18:0/22:6-PE-Pls), which are abundant especially in ascidians, are known to be crucial components for cognitive function [6,7]. Yamashita et al. conducted a study in which they orally administered PE-Pls derived from ascidians to rat models of Alzheimer’s disease and measured the reference and working memory-related learning abilities as well as the concentration of plasmalogens in organs [9]. Their results showed that the group that was administered PE-Pls derived from ascidians had relatively high concentrations of PE-Pls in plasma, erythrocytes, and the liver. In particular, the concentration of 18:0/22:6-PE-Pls in cortex, erythrocytes, and the liver was correlated with parameters of working memory-related learning ability. These findings suggest that the daily consumption of a 18:0/22:6-PE-Pl-rich diet could potentially contribute to the prevention of dementia progression. Today, research is needed to clarify the precise intake and therapeutic range of such effects. Moreover, plasmalogens are gaining attention not only for their role in dementia prevention, but also for their potential impact on other diseases and overall health. For example, plasmalogens have been suggested to have protective effects against cardiovascular disease. In a cohort study by Beyene et al., plasmalogen supplementation was shown to contribute to a reduction in the risk of type 2 diabetes, cardiovascular disease, and mortality [10]. In addition, the effects of plasmalogens on immune function have been reported, particularly in diseases associated with chronic inflammation, such as chronic fatigue syndrome, where plasmalogen supplementation has been suggested to help alleviate symptoms. A study by Navolokin et al. suggested that patients with COVID-19 and sepsis had significantly decreased levels of plasmalogens in the blood, which may contribute to the progression of chronic inflammation [5].

Plasmalogen is found in various food sources such as amberjack (muscle), ascidian (muscle, viscera), blue mussel (muscle), brown algae (leaf, sporophyll, caulome), cattle (leg muscle), chicken (leg muscle), cuttlefish (muscle, viscera), pig (leg muscle), salmon (muscle), and scallop (muscle and mantle, viscera) [11]. Previous studies have focused on the quantification of plasmalogen primarily in the muscles of marine organisms and mammals. This study focused on hen eggs as a dietary source of plasmalogens. Hen eggs are widely consumed worldwide and are known to be a rich source of lipids and proteins at a relatively low cost [12,13]. The determination of the plasmalogen content of hen eggs might contribute to the solution of plasmalogen cost and supply problems. Recent studies have suggested that the dietary intake of hen eggs might be involved in reducing the risk of dementia or improving cognitive function [14,15]. Although there have been reports on the quantification of plasmalogens in hen eggs, their contents at the molecular species level have not yet been clarified [16,17,18]. Currently, mass spectrometry (MS) and tandem mass spectrometry (MS/MS) are widely used for the quantification of lipids at the molecular species level in foodstuffs. Although MS offers high resolution and mass accuracy, it is still limited by the fact that many lipids have the same *m*/*z* values. In contrast, MS/MS can overcome this problem under certain conditions using multiple-reaction monitoring (MRM) [19]. Therefore, this study tried to determine the plasmalogen content of hen eggs at the molecular species level by electrospray ionization tandem mass spectrometry (HPLC-ESI-MS/MS), thereby clarifying the potential value of hen eggs as a dietary food that contributes to cognitive functions.

## 2. Results and Discussion

### 2.1. Fresh and Dry Weights of Hen Egg

The values of the fresh weight, dry weight, and moisture content of egg yolk, egg white, and eggshell (including the eggshell membrane) per hen egg are shown in Table 1. For fresh weights, egg white was the highest, followed by egg yolk and then eggshell. For dry weights, egg yolk was the highest, followed by eggshell and then egg white. The reversal of order between egg white and egg yolk for fresh and dry weights was due to the fact that the moisture content of egg white was almost 90 wt%.

### 2.2. Weights of Total Lipids, Neutral Lipids, Glycolipids, and Phospholipids in Hen Eggs

The values of the total lipids, neutral lipids, glycolipids, and phospholipids in egg yolk and total lipids in egg white are shown in Table 2. The total lipid content in egg yolk was high, in contrast to egg white where it was low. The highest content in egg yolk was observed for neutral lipids, followed by phospholipids and glycolipids. The total yield of these lipids was 80.6 wt%. Therefore, it is inferred that in addition to these lipids, there may be unidentified lipids (19.4 wt%) present in the total lipids of egg yolk.

### 2.3. Phospholipid Fraction of Egg Yolk

The results of the thin-layer chromatography (TLC) development of the phospholipid fraction of egg yolk along with analytical standards of phosphatidylethanolamine, phosphatidylcholine, and phosphatidylserine, are shown in Figure 1. When phospholipid fractions were developed by TLC, spots were observed for phosphatidylethanolamine, phosphatidylcholine, and phosphatidylserine. The spot for phosphatidylethanolamine was particularly large.

### 2.4. Plasmalogen Molecular Species in Egg Yolk and Egg White

Under the HPLC-ESI-MS/MS conditions used in this study, the chromatograms of each standard were obtained. With these conditions, each compound was separated in 25 min. Standard solutions of 10, 20, 50, 100, 200, 300, 400, 500, and 1,000 ng/mL were prepared and analyzed under these HPLC-ESI-MS/MS conditions, and the generated standard curves (r > 0.997) of PE-Pls are shown in Figure 2A–D, and those of PC-Pls are shown in Figure 3A–C.

HPLC-ESI-MS/MS chromatograms of the analytical standard mixture, egg yolk phospholipid fraction, and egg white total lipid extract are shown in Figure 4A–C. The results of the quantification of plasmalogen species in the egg yolk by HPLC-ESI-MS/MS are shown in Table 3 and Table 4. Three molecular species of PE-Pls, namely 18:0/18:1-PE-Pls, 18:0/20:4-PE-Pls, and 18:0/22:6-PE-Pls, were quantified. The plasmalogen species in egg yolk consisted mainly of PE-Pls, and 75.6 wt% of the quantifiable plasmalogens were 18:0/22:6-PE-Pls. The results of the quantification of plasmalogen species in the egg white by HPLC-ESI-MS/MS are shown in Table 3 and Table 4. Three molecular species of PE-Pls, namely 18:0/18:1-PE-Pls, 18:0/20:4-PE-Pls, and 18:0/22:6-PE-Pls, and one molecular species of PC-Pls, namely 18:0/20:4-PC-Pls, were quantified. Similar to egg yolk, the plasmalogen species in egg white consisted mainly of PE-Pls. Of the quantifiable plasmalogens in egg white, 49.6 wt% were 18:0/20:4-PE-Pls, and 19.2 wt% were 18:0/22:6-PE-Pls. The statistical results show a significant difference when comparing the content between yolk and egg white within the same plasmalogen species, but no significant difference was observed between different plasmalogen species (Figure 5).

### 2.5. Discussion

Plasmalogens are known to protect against neuronal death caused by factors such as oxygen and nutrient deficiency. Plasmalogens are found in marine products such as ascidians, but they are also expected to be found in common foods. In particular, hen eggs, which are rich sources of lipids and proteins, are expected to be a potential food that may have an effect on cognitive function. This study is the first quantitative assessment of hen egg plasmalogens with the molecular species level.

From the results of this study, when comparing the total plasmalogens of fresh weight in egg yolk and egg white, the egg yolk contained 1292.1 µg/100 g fresh weight, while the egg white contained 31.4 µg/100 g fresh weight. When comparing between dry weights, the egg yolk contained 2701.4 µg/100 g dry weight, while the egg white contained 279.3 µg/100 g dry weight. It was found that the total plasmalogens in egg yolk was higher than in egg white, which was 41.5 times higher in fresh weight and 9.7 times higher in dry weight. In addition, although no significant differences were observed, focusing on the absolute values, 18:0/22:6-PE-Pls were most abundant in egg yolk, and 18:0/20:4-PE-Pls were the most abundant in egg white. It became clear that the composition of plasmalogen molecular species differs between egg yolk and egg white (Figure 5).

The higher content of plasmalogens in egg yolk may be due to the different roles of egg yolk and egg white during the embryonic development process of hen eggs. Egg white is considered to be a reservoir of water and proteins, and it provides protection to the developing embryo against microorganisms by directly killing bacteria or creating an environment unfavorable for bacterial growth [20]. On the other hand, egg yolk is considered the primary nutrient source for embryonic growth, and the development of chicken embryos depends on the essential amino acids, lipids, carbohydrates, and minerals stored in the yolk [20]. Ether lipids, such as plasmalogens, have also been reported to be critical factors in cell differentiation [21]. Therefore, it can be thought that the egg yolk, which contains a higher abundance of lipids, also has a higher plasmalogen content.

Based on the results of this study, 75.6 wt% of the total plasmalogens in yolk are 18:0/22:6-PE-Pls. Among the various plasmalogen molecular species, 18:0/22:6-PE-Pls have been experimentally confirmed to be associated with concentration in the body and cognitive function [7,9]. Therefore, among egg parts, it is considered desirable to use egg yolk for the development of food products. The fresh weight of the hen eggs used in this study was 20.5 g for the egg yolk and 35.4 g for the egg white. And the combined weight of the two is 55.9 g. Based on the total plasmalogen content per egg (264.8 μg/20.5 g fresh weight for egg yolk and 11.1 μg/35.4 g fresh weight for egg white), this translates to 275.9 μg/55.9 g fresh weight. Therefore, it can be estimated that the consumption of one hen egg provides 275.9 μg (around 0.3 mg) of total plasmalogens. Since there are sources other than hen eggs that contain plasmalogens (scallops, shrimps, crabs, ascidians, and other marine products), the daily intake of plasmalogens per day in humans is considered to be higher than 1 mg via daily meals [11]. Kritz-Silverstein et al. compared the hen egg consumption and cognitive function of 617 men and 898 women over the age of 60 by the Rancho Bernardo Cohort [15]. Participants consumed between 0 and 24 eggs per week. And after more than 16 years of follow-up, it was reported that there was a positive correlation between the number of eggs consumed and performance in total recall, short-term memory, and long-term memory in men. However, no significant effects were observed in women. Although limited, their study may suggest that the consumption of over seven eggs per week contributed to improved cognitive function. According to the literature, people around the world consume approximately one hen egg per day [17,22,23,24].

Based on the results of this study, we discussed the potential that plasmalogen in egg yolk, particularly 18:0/22:6-PE-Pls, possibly contributing to cognitive function. However, some studies have also reported no significant association between egg consumption and cognitive function. For example, An et al. reported no association between egg consumption and cognitive function in older adults [25]. This discrepancy may be due to differences in study design, population studied, or the composition of the eggs consumed. The differences in egg composition may be due to the dietary components fed to the hens or differences in farming methods. On the other hand, there are concerns about the excessive consumption of eggs. Eggs are high in cholesterol. Each egg contains about 200 mg of cholesterol, which may influence the risk of cardiovascular disease. Recent studies suggest that the effect of dietary cholesterol on blood cholesterol level is not as strong as previously thought [26]. However, some studies have reported that increased egg consumption may increase the risk of cardiovascular disease in diabetic patients, so further attention is needed [27]. In the future, it may be important to determine the plasmalogen content in daily meals and develop functional foods based on such information.

As briefly mentioned in the background, eggs contain not only plasmalogens but also other components related to cognitive function. Typical ones are polar carotenoids, such as lutein and zeaxanthin [28,29]. It has been reported that there is a positive correlation between the cognitive function and concentrations of lutein and zeaxanthin in the brain, plasma, and erythrocytes of the elderly, and that these can be supplemented from daily meals [30]. Some reports have examined the combined effects of plasmalogens and carotenoids on cognitive function. For example, Takekoshi et al. reported that the co-ingestion of ascidian-derived plasmalogens and lutein-rich *Chlorella* activated brain-derived neurotrophic factor (BDNF)/tropomyosin receptor kinase B (TrkB)/cAMP response element-binding protein (CREB) signaling in the rat hippocampus [31]. Although there are reports that plasmalogen affects cognitive function, the exact mechanism has not been fully elucidated. A possible cause of discrepancies in research results related to cognitive function might be differences in the types of lipids investigated. Sidorova et al. reported that soy lecithin affects cognitive function in a different way than plasmalogen [32]. While plasmalogen supplementation improved memory and physical performance, soy lecithin increased anxiety but still improved memory and cognitive function. This highlights the importance of specifying the type of lipid in dietary interventions. Therefore, future studies should analyze and discuss not only plasmalogen in egg yolk, but also other lipids simultaneously. Furthermore, plasmalogens are not only related to cognitive function but also expected to exhibit biological activity in combination with other bioactivities. Although such considerations could not be addressed in this study, they will be essential in future research. In addition, other reports suggesting that the ingestion of hen egg components other than those described above may also potentially have a role in cognitive function [33,34,35,36]. Their reports suggest that different dietary components may influence cognitive function through different mechanisms. Moreover, there has also been interest in the physiological significance of plasmalogens in several diseases affecting cognitive function, such as sepsis, SARS-CoV-2 infection, and chronic fatigue syndrome [37,38]. Since hen eggs are composed of multiple molecules, assessing the details of their effects on cognitive function has some difficult aspects due to their complexity. Recently, advances in Artificial Intelligence (AI) technology have led to the development of new approaches that can evaluate the effects of multiple dietary components interacting to affect cognitive function [39,40]. If such AI approaches are developed in the near future, it may be possible to elucidate the more detailed effects of hen eggs on cognitive function.

Limitations of this study include potential variation in plasmalogen content between different batches of hen eggs. Plasmalogen content may vary due to uncontrolled or unreported factors such as diet, age, and environmental conditions. Further research is necessary to investigate these influences and to gain a further detailed insight into the factors that affect plasmalogen concentration. Also, the effects of storage conditions on the stability and plasmalogen content have not been thoroughly investigated. Hen eggs go through several stages of storage and handling, from production to consumption. Changes in temperature, humidity, and storage time may affect the lipid composition of hen eggs, including plasmalogens. We did not investigate the effect of cooking methods on plasmalogen contents. Previous research has shown that cooking methods can affect the nutrients in foods, and it has been reported that heating can cause the oxidation or breakdown of certain lipids [41,42]. For example, how different cooking methods such as boiling, roasting, or baking affect the stability of plasmalogens is an important point to explore in future studies. In particular, determining whether the function of plasmalogens is lost during cooking is crucial for assessing their impact on health. Studies on how these conditions affect the degradation and stability of plasmalogens are critical because they may affect the nutritional value of hen eggs.

## 3. Materials and Methods

### 3.1. Materials

Hen eggs with intensive farming were purchased from local supermarkets in Hokkaido, Japan. The producers of the purchased hen eggs were Takeuchi Poultry Farm Co., Ltd. (Hokkaido, Japan). The reagents and products used in this study were purchased from Fujifilm Wako Pure Chemical Corp., (Osaka, Japan) and were of HPLC grade or higher, unless otherwise stated. Analytical standards of each lipid used for analysis were purchased from Avanti Research, Inc. (Alabaster, AL, USA).

### 3.2. Measurement of Fresh and Dry Weights of Hen Egg

Hen eggs were separated into egg yolk, egg white, and eggshell (including the eggshell membrane) with a metal egg separator (Daiso Industries Co., Ltd., Hiroshima, Japan) and then weighed with an electronic balance (AB204-S/FACT, Mettler-Toledo International Inc., Columbus, OH, USA). These were then frozen at −30 °C in a medical refrigerator (MPR-414FR-PJ, Panasonic Corp., Osaka, Japan). After freezing, they were freeze-dried in a vacuum freeze dryer (FD-780, Tokyo Rikakikikai Co. Ltd., Tokyo, Japan) and weighed again. From these measurements, fresh weight (g) and dry weight (g) were measured, and the moisture content (wt%) was calculated. Independent quadruplicates were performed for the experiments.

### 3.3. Measurement of Total Lipid Weights of Egg Yolk and Egg White

Total lipids were extracted from the lyophilized powder of egg yolk and egg white using the Folch method, and the total lipid weight (g/100 g fresh weight) was weighed [43]. Independent quadruplicates were performed for the experiments. The eggshell was excluded from this measurement as it contained negligible amounts of lipids.

### 3.4. Measurement of Neutral Lipid, Glycolipid, Phospholipid Weights and Thin-Layer Chromatography Development of Phospholipid Fraction of Egg Yolk

The total lipids obtained (Section 3.3) were applied to a silica gel column for fractionation into neutral lipids, glycolipids, and phospholipids. Independent quadruplicates were performed for the experiments. Egg white and eggshell were excluded from this measurement due to their low total lipid content. The fractionation conditions using the silica gel column was conducted in accordance with the method described by Prasad et al. [44]. Obtained individual fractions were dried under vacuum using a rotary evaporator (N-1300, Tokyo Rikakikai Co., Ltd., Tokyo, Japan) and weighed. Then, the phospholipid fraction obtained from egg yolk was developed by TLC. Each standard of phosphatidylethanolamine, phosphatidylcholine, and phosphatidylserine was also simultaneously applied to the TLC. Silica gel (TLC silica gel 60, Merck Millipore, Danvers, MA, USA) was used for the TLC plates, and they were developed using a solvent consisting of chloroform–methanol–28 vol% ammonium hydroxide in a ratio of 65:25:5 (*v/v/v*).

### 3.5. Quantification of Plasmalogen Molecular Species in Egg Yolk and Egg White by HPLC-ESI-MS/MS

The egg yolk phospholipid fraction obtained by silica gel column fractionation (obtained from Section 3.4) and the egg white total lipids obtained from Section 3.3. were dissolved in chloroform–methanol in a ratio of 2:1 (*v/v*) to concentrations of 5 mg/mL and 1 mg/mL, respectively. These solutions were further diluted 10 or 50 times with methanol and quantified for plasmalogen molecular species by liquid chromatography coupled with HPLC-ESI-MS/MS (API 3200 triple quadrupole mass spectrometer, SCIEX, Pte. Ltd., Framingham, MA, USA). The HPLC-ESI-MS/MS system used in this study consisted of an LC-20AD HPLC pump, SIL-20AC autosampler, and CTO-20AC column oven (Shimadzu Co., Ltd., Kyoto, Japan). This was coupled to a API 3200 triple quadrupole mass spectrometer (SCIEX, Pte. Ltd., Framingham, MA, USA) equipped with electrospray ionization (ESI). This system was controlled using Analyst^®^ software ver 1.6.3. (SCIEX, Pte. Ltd., Framingham, MA, USA). The optimization procedure of each analyte was basically carried out according to the instructions of Analyst^®^ software. The multiple-reaction monitoring (MRM) transitions of each analytical standards were optimized by the MS/MS system via infusion. Then, each analytical standard was mixed to 10 μg/mL and diluted to 100 ng/mL to optimize the ion source. After that, the HPLC conditions were constructed.

The analytical conditions of the HPLC-ESI-MS/MS were adjusted and quantified according to those described by Yamashita et al. [11]. Each analytical standard of plasmalogen species (18:0/18:1-PE-Pls, 18:0/20:4-PE-Pls, 18:0/20:5-PE-Pls, 18:0/22:6-PE-Pls, 18:0/18:1-PC-Pls, 18:0/20:4-PC-Pls, 18:0/22:6-PC-Pls) with 500 ng/mL was optimized by the MS/MS system via infusion. The mass spectrometer was operated in positive ionization mode, and each plasmalogen species was detected to have a form in protonated molecules ([M+H] +). Precursor ions to the product ions (Q1 > Q3) were chosen for MRM detections of each compound. The MRMs for each compound were as follows: 18:0/18:1-PE-Pls (730 > 339), 18:0/20:4-PE-Pls (752 > 361), 18:0/20:5-PE-Pls (750 > 359), 18:0/22:6-PE-Pls (776 > 385), 18:0/18:1-PC-Pls (772 > 184), 18:0/20:4-PC-Pls (794 > 184), and 18:0/22:6-PC-Pls (818 > 184). Other parameters (desolvation potential [DP], entrance potential [EP], collision energies [CE], and collision cell exit potential [CXP]) were also optimized, and each condition is summarized in Table 5. The optimized ion source of HPLC-ESI-MS/MS using a standard mixture of 100 ng/mL resulted as follows: ionization, ESI (positive); ion source, turbo spray; collision gas (CAD), N2 (5 psi); curtain gas (CUR), N2 (20 psi); ion source gas 1 (GS1), air (50 psi); ion source gas 2 (GS2), air (50 psi); ionspray voltage (IS), 4500 V; temperature (TEM), 500 °C; and channel electron multiplier (CEM), 5500 V. The constructed HPLC conditions for the separation of each standard were follows. The column was a YMC-Pack Pro C18 (φ2.0×100mm, 5 μm (YMC Co. Ltd., Kyoto, Japan)), and the column eluent was binary gradient consisting of solvent A (water–methanol = 2:8 with 0.1% formic acid) and B (methyl tert-butyl ether–methanol = 9:1 with 0.1% formic acid). The gradient profile was as follows: 0–1 min, 0 % B (isocratic); 1–18 min, 80 % B (linear); 18–22 min, 80 % B (isocratic); 22–22.1 min, 0 % B (linear); 22.1–25 min, 0 % B (isocratic). The flow rate was adjusted to 0.2 mL/min, the injection volume was 5 uL, and the column temperature was maintained at 40 °C. The experiments were conducted as independent quadruplicates. The LOD (limit of detection) and LOQ (limit of quantification) were calculated based on Shrivastava and Gupta’s work [45], using the following formulas: LOD = 3.3σ (standard deviation)/b (slope of the regression line) and LOQ = 10σ/b (Table 5).

### 3.6. Statistical Analysis

Group statistical comparisons were conducted using Kruskal–Wallis test followed by a post hoc Steel–Dwas test for comparison between different plasmalogen species. Statistical significance was set at *p* < 0.05. For the comparison between egg yolk and egg white, Student’s *t* test was used. Statistical significance was set at *p* < 0.01. All statistical analyses were conducted using EZR (version 1.61), a software package developed at Saitama Medical University (Saitama, Japan) [46].

## 4. Conclusions

Plasmalogens are gaining attention for their potential impact on cognitive function and neuroprotection. Hen eggs have received interest as a potential source of plasmalogens. In this study, the plasmalogen content of hen eggs was quantified at the molecular species level. Our results show that both egg yolk and egg white are rich in PE-Pls. In particular, egg yolk is rich in 18:0/22:6-PE-Pls (75.6 wt% of the total plasmalogens). It was found that humans are likely to ingest almost 0.3 mg of total plasmalogens from one hen egg. These findings highlight the importance of plasmalogens in the daily diet and provide perspective for the development of new food products. In addition, this viewpoint of the benefits of hen eggs provided by this study would contribute to maintaining health and improving quality of life. Future research needs to further investigate the bioavailability and metabolism of plasmalogens in chicken eggs. In particular, focusing on how plasmalogens are absorbed and utilized in the body will be critical to better understanding their health function. In addition, investigating variations in plasmalogen content due to differences in egg factors, as well as how different egg-based dietary patterns affect cognitive function and other health benefits, will also be important research topics.

## Figures and Tables

**Figure 1 molecules-29-04795-f001:**
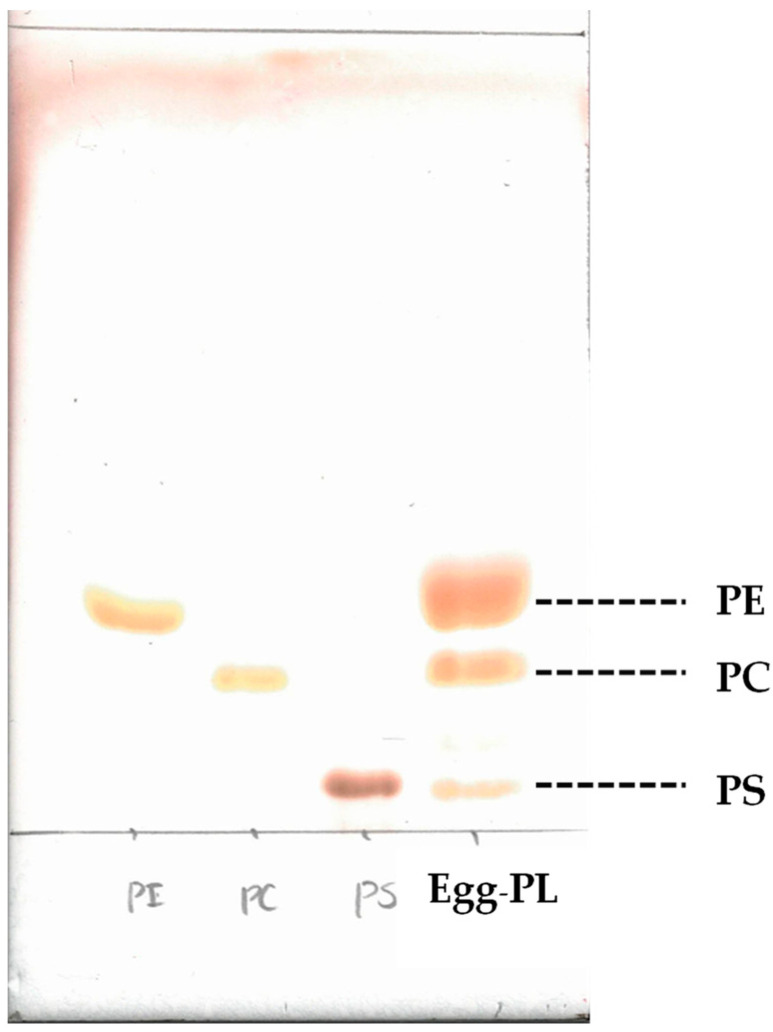
Photograph of the thin-layer chromatography development of the egg yolk phospholipid fraction. In total, 30 vol% sulfuric acid aqueous solution was used as the chromogenic reagent. Egg-PL, egg yolk phospholipid fraction; PC, phosphatidylcholine; PE, phosphatidylethanolamine; PS, phosphatidylserine.

**Figure 2 molecules-29-04795-f002:**
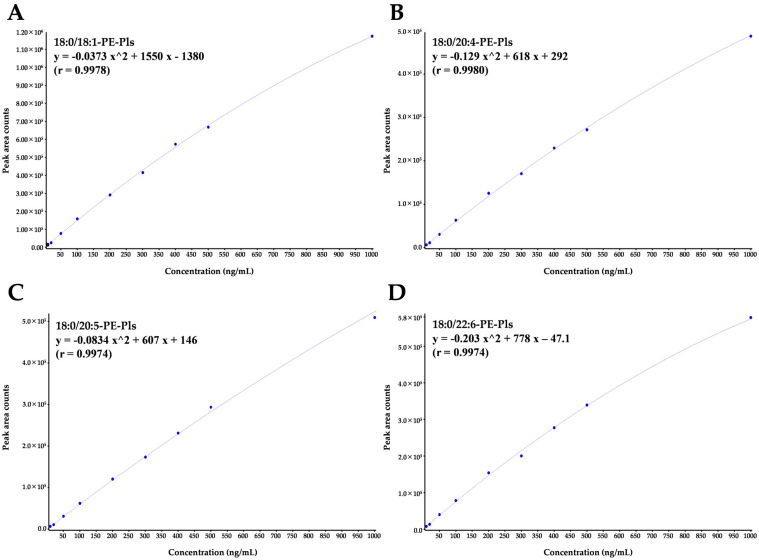
Standard curves of PE-Pls: (**A**) 18:0/18:1-PE-Pls, (**B**) 18:0/20:4-PE-Pls, (**C**) 18:0/20:5-PE-Pls, and (**D**) 18:0/22:6-PE-Pls. Standard curves were generated using Analyst^®^ software version 1.6.3 (SCIEX, Framingham, MA, USA) within a concentration range of 10 to 1000 ng/mL. MRM for each standard were as follows: 18:0/18:1-PE-Pls (730 > 339), 18:0/20:4-PE-Pls (752 > 361), 18:0/20:5-PE-Pls (750 > 359), and 18:0/22:6-PE-Pls (776 > 385). PE-Pls, ethanolamine plasmalogens.

**Figure 3 molecules-29-04795-f003:**
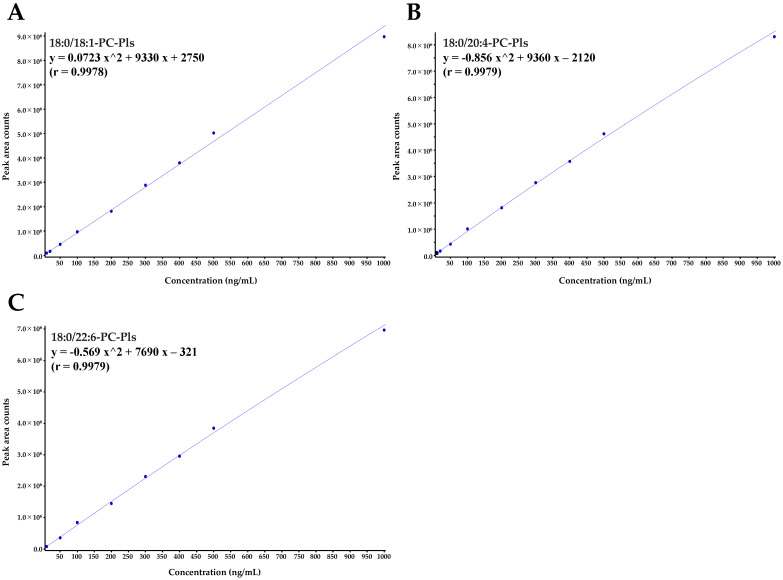
Standard curves of PC-Pls: (**A**) 18:0/18:1-PC-Pls, (**B**) 18:0/20:4-PC-Pls, and (**C**) 18:0/22:6-PC-Pls. Standard curves were generated using Analyst^®^ software version 1.6.3 (SCIEX, Framingham, MA, USA) within a concentration range of 10 to 1000 ng/mL. MRM for each standard were as follows: 18:0/18:1-PC-Pls (772 > 184), 18:0/20:4-PC-Pls (794 > 184), and 18:0/22:6-PC-Pls (818 > 184). PC-Pls, choline plasmalogens.

**Figure 4 molecules-29-04795-f004:**
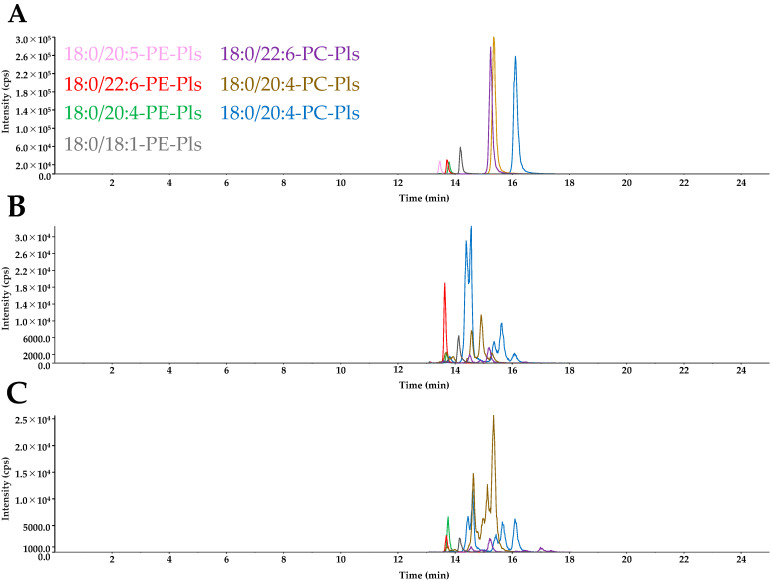
Extracted ion chromatograms from HPLC-ESI-MS/MS analysis of (**A**) analytical standard mixture (100 ng/mL), (**B**) egg yolk phospholipid fraction, and (**C**) egg white total lipid extract. MRM for each standard were as follows: 18:0/18:1-PE-Pls (730 > 339), 18:0/20:4-PE-Pls (752 > 361), 18:0/20:5-PE-Pls (750 > 359), 18:0/22:6-PE-Pls (776 > 385), 18:0/18:1-PC-Pls (772 > 184), 18:0/20:4-PC-Pls (794 > 184), 18:0/22:6-PC-Pls (818 > 184). PC-Pls, choline plasmalogens; PE-Pls, ethanolamine plasmalogens.

**Figure 5 molecules-29-04795-f005:**
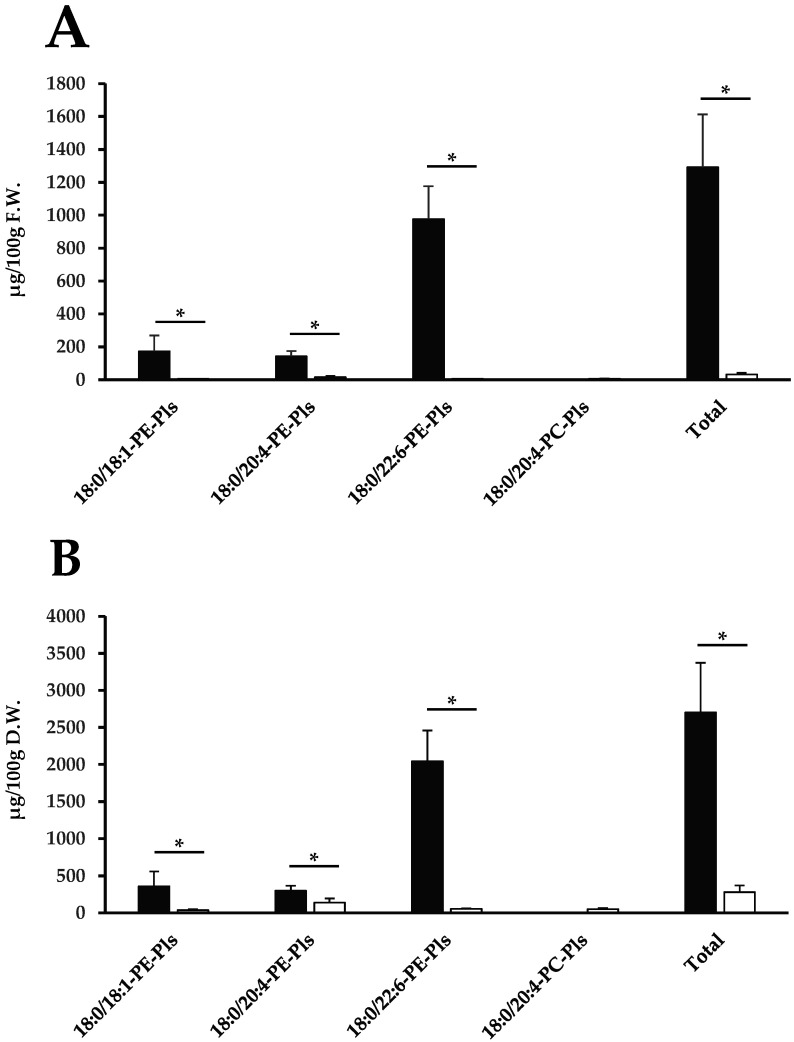
Amount of plasmalogens per 100 g of (**A**) fresh or (**B**) dried weight of egg yolk or egg white. Black bars represent egg yolk, and white bars represent egg white. For the comparison between egg yolk and egg white, Student’s *t*-test was used (* *p* < 0.01). “Total” refers to the sum of quantifiable plasmalogens (18:0/18:1-, 18:0/20:4-, and 18:0/22:6-PE-Pls for egg yolk; 18:0/18:1-, 18:0/20:4-, and 18:0/22:6-PE-Pls and 18:0/20:4-PC-Pls for egg white). Data are expressed as mean ± standard deviation (S.D.). Experiments were conducted independently in quadruplicate. Additionally, each independent sample was analyzed in technical quadruplicate. D.W., dried weight; F.W., fresh weight; PC-Pls, choline plasmalogens; PE-Pls, ethanolamine plasmalogens.

**Table 1 molecules-29-04795-t001:** Fresh weight, dry weight, and moisture content per hen egg.

	Egg Yolk	Egg White	Eggshell
Fresh weight (g/egg)	20.5 ± 1.4	35.4 ± 1.9	7.6 ± 0.2
Dry weight (g/egg)	9.8 ± 0.6	3.9 ± 0.3	6.7 ± 0.3
Moisture content (wt%)	52.2 ± 0.5	88.8 ± 0.4	12.9 ± 2.3

Data are expressed as mean ± standard deviation (S.D.). Independent quadruplicates were performed for the experiments.

**Table 2 molecules-29-04795-t002:** Weights of total lipids, neutral lipids, glycolipids, and phospholipids in hen eggs.

	Egg Yolk	Egg White
Total lipids (g/100 g D.W.)	35.5 ± 0.4	0.2 ± 0.1
Neutral lipids (g/100 g total lipid)	73.9 ± 3.4	-
Glycolipids (g/100 g total lipid)	0.5 ± 0.1	-
Phospholipids (g/100 g total lipid)	6.2 ± 0.3	-

Data are expressed as mean ± standard deviation (S.D.). Independent quadruplicates were performed for the experiments. D.W., dried weight.

**Table 3 molecules-29-04795-t003:** Amount of plasmalogens per 100 g fresh weight of egg yolk or egg white.

		Molecular Species	Value (μg/100 g F.W. of Egg Yolk or White)
Egg yolk	PE-Pls	18:0/18:1-PE-Pls	172.4 ± 95.9
		18:0/20:4-PE-Pls	142.8 ± 31.9
		18:0/22:6-PE-Pls	976.9 ± 198.8
	Total	Sum of quantifiable plasmalogens (18:0/18:1-, 18:0/20:4-, and 18:0/22:6-PE-Pls)	1292.1 ± 320.8
Egg white	PE-Pls	18:0/18:1-PE-Pls	4.1 ± 1.7
		18:0/20:4-PE-Pls	15.6 ± 6.3
		18:0/22:6-PE-Pls	6.1 ± 0.8
	PC-Pls	18:0/20:4-PC-Pls	5.6 ± 1.8
	Total	Sum of quantifiable plasmalogens (18:0/18:1-, 18:0/20:4-, 18:0/22:6-PE-Pls and 18:0/20:4-PC-Pls)	31.4 ± 10.2

Data are expressed as mean ± standard deviation (S.D.). Experiments were conducted in independent quadruplicate. Additionally, each independent sample analysis was conducted in quadruplicate as in technical quadruplicate. Molecular species not listed in the table were omitted because they were below the quantitation limit. F.W., fresh weight; PC-Pls, choline plasmalogens; PE-Pls, ethanolamine plasmalogens.

**Table 4 molecules-29-04795-t004:** Amount of plasmalogens per 100 g dried weight of egg yolk or egg white.

		Molecular Species	Value (μg/100 g D.W. of Egg Yolk or White)
Egg yolk	PE-Pls	18:0/18:1-PE-Pls	360.4 ± 200.6
		18:0/20:4-PE-Pls	298.5 ± 66.8
		18:0/22:6-PE-Pls	2042.5 ± 415.6
	Total	Sum of quantifiable plasmalogens (18:0/18:1-, 18:0/20:4-, and 18:0/22:6-PE-Pls)	2701.4 ± 670.8
Egg white	PE-Pls	18:0/18:1-PE-Pls	36.4 ± 14.9
		18:0/20:4-PE-Pls	139.1 ± 56.1
		18:0/22:6-PE-Pls	54.1 ± 7.6
	PC-Pls	18:0/20:4-PC-Pls	49.7 ± 15.8
	Total	Sum of quantifiable plasmalogens (18:0/18:1-, 18:0/20:4-, 18:0/22:6-PE-Pls and 18:0/20:4-PC-Pls)	279.3 ± 90.8

Data are expressed as mean ± standard deviation (S.D.). Experiments were conducted in independent quadruplicate. Additionally, each independent sample analysis was conducted in quadruplicate as in technical quadruplicate. Molecular species not listed in the table were omitted because they were below the quantitation limit. D.W., dried weight; PC-Pls, choline plasmalogens; PE-Pls, ethanolamine plasmalogens.

**Table 5 molecules-29-04795-t005:** Mass spectrometry parameters and LOD and LOQ of plasmalogen molecular species.

	Ionization	MRM	DP (V)	EP (V)	CE (eV)	CXP (V)	LOD (ng/mL)	LOQ (ng/mL)
18:0/18:1-PE-Pls	Positive	730 > 339	111	11	42	39	2.25	6.83
18:0/20:4-PE-Pls	Positive	752 > 361	96	11	50	35	1.60	4.84
18:0/20:5-PE-Pls	Positive	750 > 359	101	10	46	35	2.24	6.79
18:0/22:6-PE-Pls	Positive	776 > 385	111	11	46	39	1.15	3.50
18:0/18:1-PC-Pls	Positive	772 > 184	116	11	47	12	1.50	4.56
18:0/20:4-PC-Pls	Positive	794 > 184	111	11	47	12	1.08	3.27
18:0/22:6-PC-Pls	Positive	818 > 184	111	11	49	14	0.97	2.94

The volume of standard solution injected to calculate LOD and LOQ was 5 μL. CE, collision energies; CXP, collision cell exit potential; DP, desolvation potential; EP, entrance potential; LOD, limit of detection; LOQ, limit of quantification.

## Data Availability

The data that support the findings of this study are available from the corresponding authors upon reasonable request.

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
