# Peer review of "Determination of Plasmalogen Molecular Species in Hen Eggs"

_molecules, 2024, doi:10.3390/molecules29204795_

Round 1
Reviewer 1 Report
Comments and Suggestions for Authors
The research article is intriguing, but I would advise against publishing it in its current form due to some areas that require substantial improvement. Here are some specific comments:
General Comments:
- The Materials and Methods section should provide more detailed information, particularly regarding the HPLC.
- The English writing needs further improvement, as there are numerous grammatical or typographical errors throughout the manuscript. Material and Methods:
· Minor Comments:
Line 25 adds a before viewpoint
Line 41 removes the before dementia
Line 50 adds a comma before especially
Line 57 changes were to was
Line 59 changes (prevent or delay the progression) to (prevention or delay of the progression)
Line 67 changes in molecular to at the molecular
Line 67 changes has to have
Line 77 changes lipids to lipid
Line 81 changes egg to eggs
Line 91 adds the before Folch
Line 92 changes egg shell to eggshell
Line 95 adds a comma before and
Line 98 adds a comma before and also changes egg shell to eggshell
Line 99 changes its to their
Line 100 changes were to was
Line 104 adds a comma before and
Line 133 adds and before phospholipid
Line 134 adds a comma before and
Line 135 changes lipid to lipids
Line 141 adds and before phospholipid
Line 147 changes fraction to fractions
Line 161 changes Table to Tables
Line 177 changes Table to Tables
Line 172 adds a comma before and
Line 192 changes source to sources
Line 193 changes (the to affect) to (an effect)
Line 194 adds the before molecular
Line 206 adds a before concentration
Line 225 adds be after may
Line 231 adds and before zeaxanthin
Line 240 changes brain to brains
Line 241 changes On the ther to On the other
Line 251 changes have to has
Line 254 removes the before cognitive
Line 264 changes viewpoint for to the viewpoint of
Line 265 changes from to by
Comments on the Quality of English LanguageMinor editing of the English language required
Author Response
From Reviewer 1
The research article is intriguing, but I would advise against publishing it in its current form due to some areas that require substantial improvement. Here are some specific comments:
Dear Reviewer 1:
Thank you very much for reviewing this manuscript. All the comments from reviewer 1 are very important and essential for this manuscript. Thank you again for spending your valuable time on this manuscript. We made changes to this manuscript according to suggestions from reviewer 1. We have revised the points you pointed out in this revised manuscript by using Microsoft Word's revision history function. We would like you to confirm the below revised comment and revised manuscript.
Comment 1
The Materials and Methods section should provide more detailed information, particularly regarding the HPLC.
Answer:
Thank you very much for your comments. We have revised the overall explanation of the The materials and methods section. Especifically, we have added a detailed explanation of the analytical method to the revised manuscript in the section “3.5. Quantification of the plasmalogen molecular species in egg yolk and egg white by HPLC-ESI-MS/MS.” In addition, we have prapred new figure and table, “Table 5. Mass spectrometry parameters of plasmalogen molecular species,” “Figure 2. Standard curves of PE-Pls: (A) 18:0/18:1-PE-Pls, (B) 18:0/20:4-PE-Pls, (C) 18:0/20:5-PE-Pls, (D) 18:0/22:6-PE-Pls,” and “Figure 3. Standard curves of PC-Pls: (A) 18:0/18:1-PC-Pls, (B) 18:0/20:4-PC-Pls, (C) 18:0/22:6-PC-Pls.”
Comment 2
The English writing needs further improvement, as there are numerous grammatical or typographical errors throughout the manuscript.
Answer:
Thank you for carefully reviewing the English in our manuscript. We had the entire paper reviewed by someone proficient in English, and we have corrected the errors that were pointed out.
Minor Comments:
Line 25 adds a before viewpoint
Line 41 removes the before dementia
Line 50 adds a comma before especially
Line 57 changes were to was
Line 59 changes (prevent or delay the progression) to (prevention or delay of the progression)
Line 67 changes in molecular to at the molecular
Line 67 changes has to have
Line 77 changes lipids to lipid
Line 81 changes egg to eggs
Line 91 adds the before Folch
Line 92 changes egg shell to eggshell
Line 95 adds a comma before and
Line 98 adds a comma before and also changes egg shell to eggshell
Line 99 changes its to their
Line 100 changes were to was
Line 104 adds a comma before and
Line 133 adds and before phospholipid
Line 134 adds a comma before and
Line 135 changes lipid to lipids
Line 141 adds and before phospholipid
Line 147 changes fraction to fractions
Line 161 changes Table to Tables
Line 177 changes Table to Tables
Line 172 adds a comma before and
Line 192 changes source to sources
Line 193 changes (the to affect) to (an effect)
Line 194 adds the before molecular
Line 206 adds a before concentration
Line 225 adds be after may
Line 231 adds and before zeaxanthin
Line 240 changes brain to brains
Line 241 changes On the ther to On the other
Line 251 changes have to has
Line 254 removes the before cognitive
Line 264 changes viewpoint for to the viewpoint of
Line 265 changes from to by
Comments on the Quality of English Language
Minor editing of the English language required
Answer:
Thank you very much for your comments. We have made the corrections you pointed out. However, for "Line 141 adds and before phospholipid," we felt that this revision might slightly alter the intended meaning. Therefore, we made a modification according to our preference.

Reviewer 2 Report
Comments and Suggestions for Authors
The manuscript entitled "Determination of Plasmalogen Molecular Species in Hen Eggs by HPLC-ESI-MS/MS" by Taiki Miyazawa et al.
presents an interesting study on the quantification of plasmalogen molecular species in hen eggs, which is relevant for understanding dietary contributions to cognitive health. The study is well-structured, and the methods are described in detail. However, several areas require clarification and improvement.
Introduction: It would benefit from a more detailed discussion on the existing literature regarding plasmalogens in dietary sources other than hen eggs.
Materials and Methods:
- Sample Collection: The description of sample collection (hen eggs from local supermarkets) is clear. However, the manuscript should specify if there were any criteria for selecting the eggs (e.g., free-range, organic).
- Replicates and Statistical Analysis: The manuscript mentions "n=4" for various measurements. Clarify if this refers to biological or technical replicates. Additionally, provide more details on the statistical methods used to analyze the data.
Results:
- Presentation of Data: The results are presented in a clear and logical order. However, the manuscript would benefit from additional figures and tables to illustrate the data. For example, graphical representations of the plasmalogen content in egg yolk and egg white would enhance understanding.
- Interpretation of Results: The manuscript should discuss the potential reasons for the observed differences in plasmalogen content between egg yolk and egg white in more detail.
Discussion:
The discussion provides a good comparison with previous studies. However, it should address any discrepancies between this study and earlier research.
The manuscript also briefly mentions the potential health benefits of plasmalogens. This section could be expanded to discuss the implications of the findings in the context of dietary recommendations and potential therapeutic applications.
The limitations of the study are not discussed in detail. Consider addressing potential limitations, such as the variability in plasmalogen content between different batches of eggs or the influence of egg storage conditions.
Conclusion:
- Summary of Findings: The conclusion effectively summarizes the main findings. It would be beneficial to include a statement on future research directions, such as exploring the bioavailability and metabolism of dietary plasmalogens in humans.
Other Specific Comments:
1.Line 62: "This study focused on hen eggs as a dietary source of plasmalogens" – briefly mention why hen eggs were chosen over other potential sources.
3. Line 74: "Hen eggs were purchased from local supermarkets in Hokkaido, Japan" – specify if any particular type of hen egg (e.g., organic, free-range) was used.
4. Line 87: "moisture content (wt%) was calculated (n=4)" – clarify if this refers to biological or technical replicates.
5. Figure Legends: Ensure that all figures have descriptive legends that can be understood independently of the main text.
Author Response
From Reviewer 2
The manuscript entitled "Determination of Plasmalogen Molecular Species in Hen Eggs by HPLC-ESI-MS/MS" by Taiki Miyazawa et al.
Presents an interesting study on the quantification of plasmalogen molecular species in hen eggs, which is relevant for understanding dietary contributions to cognitive health. The study is well-structured, and the methods are described in detail. However, several areas require clarification and improvement.
Dear Reviewer 2:
Thank you very much for reviewing this manuscript. All the comments from reviewer 2 are very important and essential for this manuscript. Thank you again for spending your valuable time on this manuscript. We made changes to this manuscript according to suggestions from reviewer 2. We have revised the points you pointed out in this revised manuscript by using Microsoft Word's revision history function. We would like you to confirm the below revised comment and revised manuscript.
Comment 1
Introduction: It would benefit from a more detailed discussion on the existing literature regarding plasmalogens in dietary sources other than hen eggs.
Answer:
Thank you for your comments. As you pointed out, the discussion on food resources other than hen eggs was lacking in the background of this paper. In the revised manuscript, we have added and revised explanations regarding plasmalogens in dietary sources other than hen eggs and relevant information in the introduction from L73 to L81.
Comment 2
Materials and Methods:
- Sample Collection: The description of sample collection (hen eggs from local supermarkets) is clear. However, the manuscript should specify if there were any criteria for selecting the eggs (e.g., free-range, organic).
Answer:
Thank you very much. The differences in criteria for egg selection (e.g., free-range, organic farming) were not considered in this study. However, we have added the available information about the egg producers and the rearing methods in the revised manuscript in L351-353.
Comment 3
Replicates and Statistical Analysis: The manuscript mentions "n=4" for various measurements. Clarify if this refers to biological or technical replicates. Additionally, provide more details on the statistical methods used to analyze the data.
Answer:
Thank you very much. n=4 referred to biological replicates where lipids were extracted from different eggs, but the explanation was misleading. Therefore, we have added revised information in the revised manuscript at L105-106, L117-118, L209-210, L215-217, L228-229, L366, L371, L377, and L431-433. We have also added a new section “3.6. Statistical analysis.”
Comment 4
Results:- Presentation of Data: The results are presented in a clear and logical order. However, the manuscript would benefit from additional figures and tables to illustrate the data. For example, graphical representations of the plasmalogen content in egg yolk and egg white would enhance understanding.
Answer:
Thank you very much. We have created a new figure, “Figure 5. Amount of plasmalogens per 100 g of (A) fresh or (B) dried weight of egg yolk or egg white,” displaying the results of plasmalogen content as bar graphs, and added it to the revised manuscript.
Comment 5
- Interpretation of Results: The manuscript should discuss the potential reasons for the observed differences in plasmalogen content between egg yolk and egg white in more detail.
Answer:
Thank you very much for your comment. We have discussed the potential reasons for the differences in plasmalogen content between egg yolk and egg white and added this discussion to the revised manuscript in L250-259.
Comment 6
Discussion: The discussion provides a good comparison with previous studies. However, it should address any discrepancies between this study and earlier research.
Answer:
Thank you very much for your comment. We have added a more detailed explanation in the revised manuscript, in L281-295, to emphasize the discrepancies between this study and previous research.
Comment 7
The manuscript also briefly mentions the potential health benefits of plasmalogens. This section could be expanded to discuss the implications of the findings in the context of dietary recommendations and potential therapeutic applications.
Answer:
Thank you very much for your comment. We thought it would be better to add the comments you provided to the introduction rather than the discussion. Therefore, we have added the information about the potential health benefits of plasmalogens to the introduction in the revised manuscript of L62-72.
Comment 7
The limitations of the study are not discussed in detail. Consider addressing potential limitations, such as the variability in plasmalogen content between different batches of eggs or the influence of egg storage conditions.
Answer:
Thank you very much for your comment. We have added the potential limitations of this study to the revised manuscript with L329-345.
Comment 8
Conclusion:- Summary of Findings: The conclusion effectively summarizes the main findings. It would be beneficial to include a statement on future research directions, such as exploring the bioavailability and metabolism of dietary plasmalogens in humans.
Answer:
Thank you very much for your comment. A statement of future research directions has also been added to the revised paper L453-467.
Other Specific Comments:
1.Line 62: "This study focused on hen eggs as a dietary source of plasmalogens" – briefly mention why hen eggs were chosen over other potential sources.
Answer:
Thank you very much for your comment. We have added the reasons to L73-83 of the revised paper.
- Line 74: "Hen eggs were purchased from local supermarkets in Hokkaido, Japan" – specify if any particular type of hen egg (e.g., organic, free-range) was used.
Answer:
Thank you very much for your comment. We have added the available information about the egg producers and the rearing methods in the revised manuscript in L351-353.
- Line 87: "moisture content (wt%) was calculated (n=4)" – clarify if this refers to biological or technical replicates.
Answer:
Thank you very much. We have added revised information in the revised manuscript at L105-106, L117-118, L209-210, L215-217, L228-229, L366, L371, L377, and L431-433.
- Figure Legends: Ensure that all figures have descriptive legends that can be understood independently of the main text.
Answer:
Thank you very much for your comment. We have brushed up the descriptions of all figures and tables and would be happy to review the revised paper.

Reviewer 3 Report
Comments and Suggestions for Authors
This is a very brief study reporting as a novelty only the plasmalogen components of the phospholipid fraction of egg yolk and of the lipid fraction of egg white determined by HPLC-ESI-MS/MS. I would reclassify the manuscript as a "communication" rather than full length "article". There is only one figure (quite raw) and 4 tables, the last two not providing much different information.
Firstly, the manuscript has been submitted to the wrong special issue of Molecules. While the paper is about some molecules and therefore suitable for the aims and scope of this journal, the special issue is about progress in molecular spectroscopy. Please note the difference between spectroscopy and spectrometry. While the detection system of the HPLC may be an ESI tandem MS, mass spectrometry is definitely not spectroscopy (it does not make use of photons). Therefore I reject the submission of this paper to the special issue on Progress in Molecular Spectroscopy.
Here are other comments in case the paper is resubmitted to Molecules regular issue:
The Introduction is too short and too focused exclusively on the effects of plasmalogens on dementia. Is this the sole bioactivity of plasmalogens? Is there no other benefit in relation to the dietary intake of plasmalogens? Are there no concerns about them?
The same goes for the Discussion part: After a brief discussion of results, then there is an integration of other studies that cover egg consumption benefits to cognitive function. But eating too many eggs comes with some concerns also, doesn't it? What about excessive total lipid intake, cholesterol? I believe a complete discussion should put into balance the pros and cons of your "recommended" intake of eggs. What other benefits and concerns are there documented by literature? Are plasmalogens relevant exclusively for the cognitive function or are there other bioactivities associated?
Other line-by-line issues:
Lines 20 and 22: "made up" or "represented" rather than "contained"
Throughout the manuscript: et al. (there should be a full stop after "al")
Line 81: Hen eggs (plural)
Line 111: you state that "fraction developed by TLC (obtained from section 2.4)" was used for HPLC analysis. Given that you did not use preparative TLC plates as can be seen in the photograph of the plate, I doubt that you could properly scratch off the bands from the TLC plates and analyse them by HPLC. Perhaps the fraction obtained by column chromatography (before subjection to TLC development) was the one injected into the HPLC.
Line 165: you mention limit of quantitation, but nowhere in the manuscript do you report such figures of merit for the analytical chromatographic method: what are the linear range, sensitivity, LOD, LOQ, RSD for your method? Describe method validation protocols and results obtained.
Line 221: you mention the study outcome in the case of "men". What about "women"?
Author Response
From Reviewer 3
Comments and Suggestions for Authors
This is a very brief study reporting as a novelty only the plasmalogen components of the phospholipid fraction of egg yolk and of the lipid fraction of egg white determined by HPLC-ESI-MS/MS. I would reclassify the manuscript as a "communication" rather than full length "article". There is only one figure (quite raw) and 4 tables, the last two not providing much different information.
Firstly, the manuscript has been submitted to the wrong special issue of Molecules. While the paper is about some molecules and therefore suitable for the aims and scope of this journal, the special issue is about progress in molecular spectroscopy. Please note the difference between spectroscopy and spectrometry. While the detection system of the HPLC may be an ESI tandem MS, mass spectrometry is definitely not spectroscopy (it does not make use of photons). Therefore I reject the submission of this paper to the special issue on Progress in Molecular Spectroscopy.
Dear Reviewer 3:
Thank you very much for reviewing this manuscript. All the comments from reviewer 3 are very important and essential for this manuscript. Thank you again for spending your valuable time on this manuscript. We made changes to this manuscript according to suggestions from reviewer 3, as much as possible. We have revised the points you pointed out in this revised manuscript by using Microsoft Word's revision history function. We would like you to confirm the below revised comment and revised manuscript. As per your advice, we have revised the “Article” to “Communication”. We plan to consult with the journal editors and office regarding the change of the special issue for submission.
Comment 1
The Introduction is too short and too focused exclusively on the effects of plasmalogens on dementia. Is this the sole bioactivity of plasmalogens? Is there no other benefit in relation to the dietary intake of plasmalogens? Are there no concerns about them?
Answer:
Thank you very much for your comment. As you pointed out, in the introduction of this study, we only mentioned dementia in relation to plasmalogens. This is because most previous studies on plasmalogens have focused on dementia. However, recent reports have begun to address other physiological effects as well. We have added this information into L62-72 of revised manuscropt.
Comment 2
The same goes for the Discussion part: After a brief discussion of results, then there is an integration of other studies that cover egg consumption benefits to cognitive function. But eating too many eggs comes with some concerns also, doesn't it? What about excessive total lipid intake, cholesterol? I believe a complete discussion should put into balance the pros and cons of your "recommended" intake of eggs. What other benefits and concerns are there documented by literature? Are plasmalogens relevant exclusively for the cognitive function or are there other bioactivities associated?
Answer:
Thank you very much for your comment. Your comment is completely valid. We have added relevant discussions to the revised manuscript in the discussion section, lines 281-295, 312-317, and 329-345.
Other line-by-line issues:
Comment1: Lines 20 and 22: "made up" or "represented" rather than "contained"
Answer: Thank you very much for your comment. We revised into the revised manuscript.
Comment2: Throughout the manuscript: et al. (there should be a full stop after "al")
Answer: Thank you very much for your comment. We revised into the revised manuscript.
Comment3: Line 81: Hen eggs (plural)
Answer: Thank you very much for your comment. We revised into the revised manuscript.
Comment4: Line 111: you state that "fraction developed by TLC (obtained from section 2.4)" was used for HPLC analysis. Given that you did not use preparative TLC plates as can be seen in the photograph of the plate, I doubt that you could properly scratch off the bands from the TLC plates and analyse them by HPLC. Perhaps the fraction obtained by column chromatography (before subjection to TLC development) was the one injected into the HPLC.
Answer: Thank you very much for your important feedback. As you pointed out, we made an error in the description. We have added the corrected information to the revised manuscript in the section “3.5. Quantification of the plasmalogen molecular species in egg yolk and egg white by HPLC-ESI-MS/MS.”
Comment5: Line 165: you mention limit of quantitation, but nowhere in the manuscript do you report such figures of merit for the analytical chromatographic method: what are the linear range, sensitivity, LOD, LOQ, RSD for your method? Describe method validation protocols and results obtained.
Answer: Thank you very much for your comment. We were not able to evaluate the linear range, sensitivity, LOD, LOQ, and RSD in this study. However, we have provided as much detail as possible regarding the chromatographs, analytical methods, and results, and have added new figures and tables.
Comment6:Line 221: you mention the study outcome in the case of "men". What about "women"?
Answer: Thank you very much for your comment. This reference reported no effect in Women, so we have revised it in the revised manuscript L274-279.

Reviewer 4 Report
Comments and Suggestions for Authors
The manuscript “Determination of plasmalogen molecular species in hen eggs by HPLC-ESI-MS/MS” discusses the determination of plasmalogen molecular species in hen eggs using HPLC-ESI-MS/MS. The study explores the concentrations of plasmalogens in egg yolk and egg white, highlighting the higher content in egg yolk. The manuscript is in the aim and scope of the Molecules journal. However, it needs to be polished, and some meaningful parts are missing. For these reasons, I suggest a major revision.
From the title, the study must be focused on the determination of plasmalogen molecular species in eggs with an HPLC MS/MS. However, I do not understand the dementia topic. It would be more concise and in topic to speak if there are other HPLC MS/MS methods for the determination of plasmalogen in eggs, why use HPLC and why mass spectrometry. The introduction part must be rewritten.
L81: How were separated the hen egg and egg yolk? How many hen eggs were analysed?
L101-L102: Which model of rotary evaporator was used?
L110-L118: The method was validated? Which analytical parameters were determined/used for the HPLC-ESI-MS/MS method? Even if you use the Yamashita et al. method, you have to test the reproducibility of the method also on your instrument. Which were the LOQ and the LOD? The repeatability? The recovery? Please, I want to see the chromatograms of blank samples, fortified samples (if used), real samples, and the correlation coefficient and the equations of the calibration curve.
The data were not statistically analysed. There is no “statistical analysis were conducted”. Therefore the data are not meaningful without a statistical analysis. You have to test the differences with the statistical analysis and with the correct software. For these reasons, the Results and the discussion part must be rewritten according to the results of the statistical analysis.
L164-L165: “were below the limit of quantification and are therefore presented only as a reference value”. Do you have any references about that way to process the under LOQ values?
It could be an interesting addition to your study to include an analysis of the content of egg plasmalogens in eggs cooked using different methods. This would add an innovative and forward-thinking aspect to your research.
The discussion part is not clear because you discuss the data and the content of other researchers and not yours. L228-248 is not about the HPLC MS/MS method nor about plasmalogens. So, you have to remove it.
Some parts of the paper are a Review, and some are an Article. Uniform is the way of describing. If you want to analyse the correlation between dementia and egg consumption, do a review; if you want to analyse plasmalogens in eggs, focus on it. Improve the analytical method, showing chromatograms, analytical parameters, and recoveries. The focus of the manuscript needs to be clarified.
Author Response
From Reviewer 4
The manuscript “Determination of plasmalogen molecular species in hen eggs by HPLC-ESI-MS/MS” discusses the determination of plasmalogen molecular species in hen eggs using HPLC-ESI-MS/MS. The study explores the concentrations of plasmalogens in egg yolk and egg white, highlighting the higher content in egg yolk. The manuscript is in the aim and scope of the Molecules journal. However, it needs to be polished, and some meaningful parts are missing. For these reasons, I suggest a major revision.
Dear Reviewer 4:
Thank you very much for reviewing this manuscript. All the comments from reviewer 4 are very important and essential for this manuscript. Thank you again for spending your valuable time on this manuscript. We made changes to this manuscript according to suggestions from reviewer 4, as much as possible. We have revised the points you pointed out in this revised manuscript by using Microsoft Word's revision history function. We would like you to confirm the below revised comment and revised manuscript.
Comment 1
From the title, the study must be focused on the determination of plasmalogen molecular species in eggs with an HPLC MS/MS. However, I do not understand the dementia topic. It would be more concise and in topic to speak if there are other HPLC MS/MS methods for the determination of plasmalogen in eggs, why use HPLC and why mass spectrometry. The introduction part must be rewritten.
Answer:
Thank you very much for your comment. We have rewritten the introductory section, and we would appreciate it if you could review the revised manuscript. As you suggested, we have added an explanation, particularly regarding the reason for using mass spectrometry. We were unsure whether to remove the part about dementia, but since a significant amount of research on plasmalogens focuses on dementia, and other reviewers had recommended including it in the background, we decided to keep it.
Comment 2
L81: How were separated the hen egg and egg yolk? How many hen eggs were analysed?
Answer:
Thank you very much for your comment. n=4 referred to biological replicates where lipids were extracted from different eggs, but the explanation was misleading. Therefore, we have added revised information in the revised manuscript at L105-106, L117-118, L209-210, L215-217, L228-229, L366, L371, L377, and L431-433.
Comment 3
L101-L102: Which model of rotary evaporator was used?
Answer:
Thank you very much for your comment. We wdded the model of rotary evaporator into the L381-382 of revised manuscript.
Comment 4
L110-L118: The method was validated? Which analytical parameters were determined/used for the HPLC-ESI-MS/MS method? Even if you use the Yamashita et al. method, you have to test the reproducibility of the method also on your instrument. Which were the LOQ and the LOD? The repeatability? The recovery? Please, I want to see the chromatograms of blank samples, fortified samples (if used), real samples, and the correlation coefficient and the equations of the calibration curve.
Answer:
Thank you very much for your comment. We were not able to evaluate the linear range, sensitivity, LOD, LOQ, and RSD in this study. However, we have provided as much detail as possible regarding the chromatographs, analytical methods, and results, and have added new figures and tables.
Comment 5
The data were not statistically analysed. There is no “statistical analysis were conducted”. Therefore the data are not meaningful without a statistical analysis. You have to test the differences with the statistical analysis and with the correct software. For these reasons, the Results and the discussion part must be rewritten according to the results of the statistical analysis.
Answer:
Thank you very much for your comment. As you pointed out, we had not adequately interpreted the experimental results based on the statistical analysis. We have added a new explanation in the “3.6. Statistical analysis” section of the Material and Methods, and we also created a new Figure 5. These results were also discussed in the discussion section.
Comment 6
L164-L165: “were below the limit of quantification and are therefore presented only as a reference value”. Do you have any references about that way to process the under LOQ values?
Answer:
Thank you very much for your comment. In this study, samples that did not fall within the range of the calibration curve were considered to be below the limit of quantification, but were calculated by Analyst® software ver. 1.6.3 as a reference value. We added these information into L433-435 of revised manuscript.
Comment 7
It could be an interesting addition to your study to include an analysis of the content of egg plasmalogens in eggs cooked using different methods. This would add an innovative and forward-thinking aspect to your research.
Answer:
Thank you very much for your comment. As you pointed out, it is indeed very important to consider how cooking methods affect the content of components in hen eggs. We have added this as a direction for future research in the discussion section, lines 320-345, of the revised manuscript. Additionally, since we were not able to conduct experiments on this aspect, we have decided to submit this paper as a "Communication" rather than an "Article".
Comment 8
The discussion part is not clear because you discuss the data and the content of other researchers and not yours. L228-248 is not about the HPLC MS/MS method nor about plasmalogens. So, you have to remove it.
Answer:
Thank you very much for pointing this out. We wondered whether or not to delete the relevant discussion, but some comments recommended this content, so after much deliberation we decided to leave it. However, I understand that you have a valid point, and I have revised the entire discussion and would be happy for you to review it. We also decided to remove the words HPLC-MS/MS in the title together.
Comment 9
Some parts of the paper are a Review, and some are an Article. Uniform is the way of describing. If you want to analyse the correlation between dementia and egg consumption, do a review; if you want to analyse plasmalogens in eggs, focus on it. Improve the analytical method, showing chromatograms, analytical parameters, and recoveries. The focus of the manuscript needs to be clarified.
Answer:
Thank you very much. We have improved the discussion of the revised manuscript so that, whenever possible, it is presented as a discussion of the research article rather than a review.

Reviewer 5 Report
Comments and Suggestions for Authors
In this study, the authors determined the concentrations of plasmalogen molecular species in hen eggs using a previously developed LC-MS procedure (ref. 19). Due to the need to find potential dietary sources of plasmalogen of people with dementia, the study is of significance from a public health perspective. In general the study design is logical and presented data are of acceptable quality. Below please find a few minor comment for consideration.
1. The authors need to revise the abstract to improve readability, especially why dietary intake of plasmalogen would be a promising remedy for dementia.
2. Quality of Figure is poor.
3. LC-MS method. Any modifications from the previous method (Ref. 19). if so the authors need to provide the experimental details and justifications.
4. Tables 3 and 4. Report relative standard derivations (RSDs) as well. There are measurements with large variability (e.g., Table 3. 18:0/18:1-PE-Pls 193.8 ± 116.7 and Table 4. 18:0/18:1-PE-Pls 405.1 ± 243.8). For these measurements, the RSDs are greater than 50%. The authors need to explain the sources of uncertainty in the discussion. Any QC samples included in the course of analysis? If so the data should be provided.
Comments on the Quality of English LanguageMinor editing of English language required.
Author Response
From Reviewer 5
In this study, the authors determined the concentrations of plasmalogen molecular species in hen eggs using a previously developed LC-MS procedure (ref. 19). Due to the need to find potential dietary sources of plasmalogen of people with dementia, the study is of significance from a public health perspective. In general the study design is logical and presented data are of acceptable quality. Below please find a few minor comment for consideration.
Dear Reviewer 5:
Thank you very much for reviewing this manuscript. All the comments from reviewer 5 are very important and essential for this manuscript. Thank you again for spending your valuable time on this manuscript. We made changes to this manuscript according to suggestions from reviewer 5. We have revised the points you pointed out in this revised manuscript by using Microsoft Word's revision history function. We would like you to confirm the below revised comment and revised manuscript.
Comment 1
- The authors need to revise the abstract to improve readability, especially why dietary intake of plasmalogen would be a promising remedy for dementia.
Answer:
Thank you very much for your comment. Your opinion is plausible. We have revised the abstract to make the significance of this study easier to understand.
Comment 2
- Quality of Figure is poor.
Answer:
Thank you very much for your comment. We have revised the figures. Additionally, we have created new figures: "Figure 2. Standard curves of PE-Pls: (A) 18:0/18:1-PE-Pls, (B) 18:0/20:4-PE-Pls, (C) 18:0/20:5-PE-Pls, (D) 18:0/22:6-PE-Pls," "Figure 3. Standard curves of PC-Pls: (A) 18:0/18:1-PC-Pls, (B) 18:0/20:4-PC-Pls, (C) 18:0/22:6-PC-Pls," "Figure 4. Extracted ion chromatograms from HPLC-ESI-MS/MS analysis," and "Figure 5. Amount of plasmalogens per 100 g of (A) fresh or (B) dried weight of egg yolk or egg white." We have included these in the revised manuscript, and we would appreciate it if you could review them.
Comment 3
- LC-MS method. Any modifications from the previous method (Ref. 19). if so the authors need to provide the experimental details and justifications.
Answer:
Thank you very much for your comment. We have revised the overall explanation of the The materials and methods section. Especifically, we have added a detailed explanation of the analytical method to the revised manuscript in the section “3.5. Quantification of the plasmalogen molecular species in egg yolk and egg white by HPLC-ESI-MS/MS.” In addition, we have prapred new figure and table, “Table 5. Mass spectrometry parameters of plasmalogen molecular species,” “Figure 2. Standard curves of PE-Pls: (A) 18:0/18:1-PE-Pls, (B) 18:0/20:4-PE-Pls, (C) 18:0/20:5-PE-Pls, (D) 18:0/22:6-PE-Pls,” and “Figure 3. Standard curves of PC-Pls: (A) 18:0/18:1-PC-Pls, (B) 18:0/20:4-PC-Pls, (C) 18:0/22:6-PC-Pls.”
Comment 4
Tables 3 and 4. Report relative standard derivations (RSDs) as well. There are measurements with large variability (e.g., Table 3. 18:0/18:1-PE-Pls 193.8 ± 116.7 and Table 4. 18:0/18:1-PE-Pls 405.1 ± 243.8). For these measurements, the RSDs are greater than 50%. The authors need to explain the sources of uncertainty in the discussion. Any QC samples included in the course of analysis? If so the data should be provided.
Answer:
Thank you very much. We have included a discussion of the causes of the variation in the hen egg experimental data in the discussion. Additionally, since we were not able to added the RSDs, we have decided to submit this paper as a "Communication" rather than an "Article".
Comment 5
Minor editing of English language required.
Answer:
Thank you very much. A reader who is fluent in English pointed out problem areas in the English language, so we have corrected various parts of the text. We would be happy if you could check the revised manuscript.

Round 2
Reviewer 3 Report
Comments and Suggestions for Authors
No further comments
Author Response
Dear Reviewer3,
We would like to express our sincere gratitude for your thorough review of this manuscript. Your valuable comments were essential in improving the quality of the manuscript.
Reviewer 4 Report
Comments and Suggestions for Authors
The authors have made significant improvements to the manuscript in response to my comments, resulting in a considerable enhancement of the overall quality.
However, the authors' response indicates that the method has not been validated for the LOD, LOQ, and linearity parameters. Given the critical importance of reproducibility and the credibility of analytical data, it is essential that these parameters are properly validated. In my opinion a manuscript presenting analytical results without established LOD and LOQ values cannot be considered acceptable for publication.
I suggest to improve the analytical method because the authors and the research topic is really interesting.
Author Response
Comment from Reviewer 4:
The authors have made significant improvements to the manuscript in response to my comments, resulting in a considerable enhancement of the overall quality.
However, the authors' response indicates that the method has not been validated for the LOD, LOQ, and linearity parameters. Given the critical importance of reproducibility and the credibility of analytical data, it is essential that these parameters are properly validated. In my opinion a manuscript presenting analytical results without established LOD and LOQ values cannot be considered acceptable for publication.
I suggest to improve the analytical method because the authors and the research topic is really interesting.
Dear Reviewer 4:
We would like to express our sincere gratitude for your thorough review of this manuscript. Your valuable comments were essential in improving the quality of the manuscript. In consideration of your comments, we have added the LOD and LOQ values information to the Methods section of the revised manuscript and to TABLE 5, which we would appreciate your review. We have also revised the entire text to reflect your and the editor's comments, reporting all compounds that are not within the calibration range as “not quantifiable”.
Thanks again for your comments, they are much appreciated.